# A Comprehensive Study for Determination of Free Fatty Acids in Selected Biological Materials: A Review

**DOI:** 10.3390/foods13121891

**Published:** 2024-06-16

**Authors:** Beyza Uçar, Zahra Gholami, Kateřina Svobodová, Ivana Hradecká, Vladimír Hönig

**Affiliations:** 1ORLEN UniCRE a.s., Revoluční 1521/84, 400 01 Ústí nad Labem, Czech Republic; zahra.gholami@orlenunicre.cz (Z.G.); ivana.hradecka@orlenunicre.cz (I.H.); 2Department of Chemistry, Faculty of Agrobiology, Food and Natural Resources, Czech University of Life Sciences Prague, Kamýcká 129, 165 00 Prague, Czech Republic; honig@af.czu.cz

**Keywords:** FFA analysis, electrochemical methods, colorimetric methods, spectroscopic methods, chromatographic methods introduction

## Abstract

The quality of oil is highly dependent on its free fatty acid (FFA) content, especially due to increased restrictions on renewable fuels. As a result, there has been a growing interest in free fatty acid determination methods over the last few decades. While various standard methods are currently available, such as the American Oil Chemists Society (AOCS), International Union of Pure and Applied Chemistry (IUPAC), and Japan Oil Chemists’ Society (JOCS), to obtain accurate results, there is a pressing need to investigate a fast, accurate, feasible, and eco-friendly methodology for determining FFA in biological materials. This is owing to inadequate characteristics of the methods, such as solvent consumption and reproducibility, among others. This study aims to investigate FFA determination methods to identify suitable approaches and introduce a fresh perspective.

## 1. Introduction

There is an increasing interest in oil sources due to the high energy demand for replacing fossil fuels. International agreements, such as RED directives, the Paris protocol, and the International Civil Aviation Organization (ICAO), encourage sustainable eco-friendly fuel investigations due to increased restrictions [1,2,3,4,5,6]. Moreover, used cooking oil (UCO), vegetable oil (VO), and animal fats are considered as renewable feedstock in the fuel industry to reduce greenhouse gas emissions (GHG) [6,7,8,9,10]. These materials provide high biodegradability, thermal stability, and low toxicity for the environment. However, the quality and composition of oils are critical factors, and they are not suitable for direct usage due to their high viscosity, engine oil contamination, etc. [9,11].

In the overall composition of VOs and UCO (Figure 1), the main ingredients consist of free fatty acids (FFAs) and triglycerides (TG), while minor components exist in approximately 1–5%. These minor components have significant effects on the material quality due to the presence of metals, fatty alcohols, oxidation products, and others. Especially, the concentration of carboxylic acids, such as naphthenic acids in oil, significantly influences corrosion in the engine [12]. Moreover, the composition of VOs and UCO possibly contains variable levels of moisture, insoluble impurities, unsaponifiable compounds, and some lipid oxidation compounds, which can influence their energy value [13]. The quality of these oils is primarily influenced by applied processes, storage conditions, and frying temperatures [14,15,16]. In several countries, the permissible frying temperature range is legally set between 170 and 180 °C, and the legal FFA limit is between 0 and 2.5% in the frying oil due to oxidation, hydrolysis, and thermal degradation processes [17,18,19]. Increased viscosity and color change of edible oils are observed during frying due to being exposed to high temperatures for a long time as a result of these chemical processes [20,21]. As a result of the increased FFA content, the oil requires vast purification steps to be utilized as a valuable feedstock in fuel production.

The FFA content is measured with the regulation No. 2568/91 method in many countries [17,22,23,24]. The valid analytical method should have sufficient selectivity, precision, reproducibility, accuracy, linear range, limit of detection/quantitation, recovery, robustness, and stability for FFA detection [25]. Standard methods are developed by organizations such as the American Oil Chemists Society (AOCS), International Union of Pure and Applied Chemistry (IUPAC), and Japan Oil Chemists’ Society (JOCS) [15,26,27]. Among these methods, the titration method is the most common and the oldest FFA determination technique for VOs and UCO [15,26,28]. However, this method suffers from lower accuracy and relatively high solvent consumption, prompting the need for research into alternative methods, such as thermometric, calorimetric, electrochemical, spectroscopic, and chromatographic methods.

The thermometric titration method is an altered titration method that utilizes a sensor to optimize solvent consumption (isopropanol, NaOH solution, n-butylamine, etc.) and can also be performed on colored samples via optimization [29,30,31]. The thermometric titration method exhibits the total acid value of the applied sample. Nonetheless, this method is one of the oldest acid value determination methods, so it has been tested for different detection purposes, including measuring water content as an alternative to Karl Fisher titration [32]. Although it provides reliable outcomes, the issue of significant solvent consumption and the need for detailed information persists. Thus, the development of alternative methods is encouraged.

Calorimetric methods, such as copper soap and flow injection (FI), create an alternative approach to the classical titration method. The copper soap method, which is widely used in quantitative kinetic studies, measures the color intensity by occurring copper soaps of FFAs with the cupric acetate reagent [33]. The FI method is helpful to control solvent consumption during analysis, and it can also be coupled with other methods to obtain a high prediction efficiency, such as UV, mass spectroscopy (MS), the voltametric method, etc. In the FI method, the analyzed samples are diluted off-line or on-line with organic solvents, such as methanol, ethanol, propanol, toluene, xylene, or hexane [34].

Electrochemical methods are classified as voltametric, electrical conductivity (EC), and pH metric methods. Voltametric method is generally convenient to determine heavy metals in the analyzed sample. However, it is useful to detect FFA content by coupling with FI, as shown in previous research [22,35]. EC measures the conduction capability of a material existing in an electric current by utilizing concentrations of anions and cations. Thus, the system produces accurate FFA data independent of the oil type [36]. The pH metric method is performed without titration to determine the acid value (AV) by employing the reagent of triethanolamine (TEA) in potassium nitrate (KNO_3_), water, and alcohol (ethanol or isopropanol (i-PrOH)) [37]. These electrochemical methods have advantages such as low cost, simplicity, and high reproducibility, although they require high solvent consumption. Nevertheless, thermometric, calorimetric, and EC methods demonstrate overall FFA values, which can result in limitations in terms of accuracy, reproducibility, and a lack of detailed information regarding specific chemical compounds of the applied samples.

Conversely, spectroscopic methods are helpful tools to analytically determine the full skeletal structure of the materials. Infrared and Raman methods gained attention for their non-destructiveness, high accuracy, high sensitivity, and direct results, based on either transmission or reflection measurements [38,39,40,41,42]. Both of the spectroscopic methods utilize molecular vibrations with different pathways, corresponding to changes in polarizability as intense Raman bands or intense infrared absorption bands [43]. Regarding the latter, nuclear magnetic resonance (NMR) spectroscopy is a relatively new innovative analytical methodology with high sensitivity, accuracy, rapidity, and resolution. It requires high equipment costs, demonstrating a disadvantage [44]. All spectroscopic techniques are suitable to clearly identify the composition of various oils and their FFA content.

Additionally, chromatographic methods provide reliable predictions to create a reference for infrared methods [45]. The most common methods are gas chromatography (GC) and high-performance liquid chromatography (HPLC) for FFA determination of VOs and UCO [46]. HPLC is generally utilized to detect non-volatile, high-molecular-mass components, coupling with adsorption/partition chromatography [47]. GC can be applied on volatile samples to identify the content and skeletal structure of the sample [48]. GC-MS is the most utilized GC method for FFA detection, and it exhibits high efficiency regarding the utilized carrier gas, oven temperature, and column characteristics during the analysis [49].

The determination of FFAs, peroxides (POs), fatty acid profile, total polar components (TPC), and carbonyl compounds is essential to apply suitable pretreatment methods to VOs and UCO for the purpose of renewable fuel production [22]. From this perspective, this review aims to provide in-depth analysis of the methods used to determine fatty acids in biological materials, focusing on advanced instrumental analytical techniques. By expanding the scope to include alternative methods, this review seeks to offer a comprehensive guide for researchers and industry professionals in selecting the most appropriate methodologies for fatty acid analysis.

## 2. FFA Analysis Methods

Many methods have been investigated and tested in various studies to respond to industrial concerns for FFA determination. The most common methods are listed in Figure 2.

### 2.1. Classical Titration Methods

In the classic titration method, the acid value of VOs is determined by utilizing the color change during alkaline titration: (1) potassium hydroxide (KOH) or sodium hydroxide (NaOH) solutions mixed with an oil sample, or (2) oil dissolved in ether-alcohol solution, and phenolphthalein utilized as an indicator (AOCS) to determine the FFA value [50,51]. The end point, the required volume of solvent consumption, is determined by the obtained stable color change. Total acidity is determined by the equation presented below [16,38]:(1)$$\%\;FFA(\%\;m/m)=\frac{V\times M\times 28.2}{m}$$
where “% FFA” is the percentage of free fatty acids, “V” represents the volume of the consumed solvent, “M” denotes the molarity of the NaOH solution, and the mass of the oil sample is shown by “m”.

In the research of Li et al. [50], acid values of commercial vegetable oils were tested with the classical titration method. The results were accurate, with high chemical consumption of each sample (3–4 g) testes [50]. Moreover, Barthet et al. [26] performed analysis tests for marine and vegetable oils, and presented that this method was not suitable for acid determination of large amounts of sample (e.g., 56 g) due to the unformed stable color change. Also, they mentioned that when the highly colored oil was tested, 7 g of oil sample was required, according to AOCS restrictions [26].

Since the classical titration method required high amounts of solvent consumption, Anconi et al. [52] proposed a method conducted with a smartphone camera using the classical titration method. They tested extra virgin oil and various VOs for free acid determination. This altered titration method was observed to lead to less sample requirement (e.g., 0.5 g), less chemical consumption, and a total ~2 min test time, with R^2^ of 98 and a mean absolute error of 0.06% without the usage of any special equipment [52]. Their research demonstrated that the classical method might be improved. Nevertheless, it suffered from presenting inadequate FFA characteristics, textural properties of tested materials, and limitations for testing colored samples.

### 2.2. Instrumental Methods

#### 2.2.1. Colorimetric Methods

##### Copper Soap Method

The copper soap method is a calorimetric method. In the methodology, oil (~8 g) and standard alkali (~30 mL) are titrated to determine the acid value, which is called “saponification” [53,54]. Cu(NO_3_)_2_.3H_2_O and cupric acetate–pyridine are likely to be employed as the copper reagent [55]. Via extraction of the aqueous solution (~5 min), copper salts of the fatty acids are formed as blue colored. The intensity of the color changes regarding the FFA concentration, which is measured by a colorimeter or spectrophotometer. When the % FFA increases, the % transmittance becomes lower owing to the intense color change [53]. These copper soaps are highly soluble in organic solvents, such as benzene, chlorobenzene, xylene, 1-propanol, 1-butanol, and 1-pentanol, and they may have a corrosive impact in the engines, which necessitates a protective layer [54,56]. Nevertheless, the copper soaps can be used in cosmetic and pharmaceutical applications due to their antibacterial properties [54]. In a similar perspective, Bhutra et al. [57] investigated biomedical use of copper soaps, which was derived from the FFA analysis of VO and UCO. They observed that fungi toxicity increased with the increased FFA value of the oil sample, which was promising for agricultural protection [57]. Thus, copper soaps may contribute to valuable chemical production instead of being preserved as waste.

In the copper soap method, the utilized solvent amount was relatively high and the sampling frequency was lower than the classical titration method [15,53,58]. Thus, this method does not provide an eco-friendly approach nor in-depth information on FFA compounds.

##### Flow Injection Analysis (FIA) Methods

The FIA method has been used for determining the FFA content of variable oils via an automation system [15,58]. This method is based on monitoring the neutralization in the absorbance of the indicators, which can be phenolphthalein (PHP), bromothymol blue (BTB), or KOH, by the injected sample. Sample throughputs are variable regarding utilized FIA, which can be 35–74 h^−1^ for single-line FIA and 15–40 h^−1^ for two-line FIA. It is a rapid determination system and can be modified with HPLC. For the latter, application of the derivatization method may offer higher sensitivity and accuracy of the outcomes [15,59,60]. As an example of this combination, Ayyildiz et al. [61] investigated FFA determination of corn oil via FIA conducted with HPLC. They proposed more eco-friendly FFA detection of oil samples with the amounts of 1.6 mL of n-propanol, 1 μL of oil sample, and 30 μL of reagent (KOH and PHP solutions) per analysis (75 h^−1^). They also confirmed that the results presented better sensitivity and accuracy than the reference FIA method while preserving the inherent properties [61].

Moreover, the FIA method presents versatility, simplicity, low cost, high precision, reproducibility, and low solvent consumption as advantages by optimizing the operating conditions, such as pH, flow rate, tubing length/diameter, and proportion of oil to solvent. However, there are obstacles, such as reagent dilution, mixing, and flow, to controlling the conditions and obtaining high stability [62]. This method did not gain much attention due to its stability problem. Thus, this method might be preferable when instability issues are overcome.

#### 2.2.2. Electrochemical Methods

##### Voltametric Methods

As an alternative to other methods, voltametric methods can be applied to determine FFA content in oils. This method presents less solvent usage than the classic titration method. Heavy and toxic metals can be detected with various types of voltametric methods, such as adodic stripping voltammetry (ASV). Nevertheless, the voltametric approach is rarely applied to direct FFA detection due to the poor conductivity of the matrix, although it has a wide usage area, especially in analytical chemistry. This can be solved by utilizing tri-hexyl (tetradecyl)phosphonium bis (trifluoromethylsulfonyl) imide ([P_14,6,6,6_]^+^[NTf_2_]^−^) as a supporting electrolyte [22,23,58,63].

The typical sweep technique (cyclic voltammetry) includes a three-electrode cell (Figure 3). Pt and Au microelectrodes in [P_14,6,6,6_]^+^[NTf_2_]^−^ solution contain electroactive species under nitrogen- or argon-purged conditions [22,23,63]. Besides, a quinone reagent (2-methyl-1,4-naphthoquinone (VK3)), LiClO_4_ in ethanol solution and diphenylphosphinoethane ([CoCp(dppe)(CH_3_CN)](PF_6_)_2_), can be used as a supporting electrolyte instead of Pt/Au [P_14,6,6,6_]^+^[NTf_2_]^−^ solution, as Li S. G. et al. [50] and Elgrishi et al. [64] utilized, respectively. The data range can be achieved from −2 V to +2 V to determine the FFA content via the cationic reduction process [22,23,63].

Electrochemical reduction of quinone in amphiprotic solvents presents great accuracy in a short time and fixed potential coupling with HPLC for determining FFA. By the usage of this improvement, it is possible to obtain sample throughputs within 20 min [50,65].

Although the voltammetry method requires less solvent usage and can be coupled with the flow injection system for better sensitivity and reproducibility, it is not applicable for viscous samples owing to overlapping and feasibility problems [23,58,63].

##### Electrical Conductivity (EC) Methods

The electrical conductivity (EC) method is one of the alternative methods, and it may be preferable mainly thanks to the low equipment cost. The methodology is based on recording changes in the EC value regarding the ability of a material to conduct an electric current to determine the FFA value of the tested material [7,36]. Yu et al. [36] tested various edible oils using the EC method. They utilized KOH solution, which supplied the required anions and cations (K^+^ and OH^−^), to predict EC value changes, and this KOH solution was supposed to be at a low concentration (~0.04 M) to prevent the saponification of the oil sample (4 g). The oil sample was dispersed into a 40 mL KOH solution, and the overall procedure took approximately 5 min. Their results were accurate with a high amount of solvent usage [36]. Besides employment of the KOH solution, Yang et al. [7] investigated applying potassium iodide (KI; 1 mL) as an anion/cation source for the analysis of edible oils via the EC method, and the results were accurate, with R = 0.99 [7]. In this way, the declined solvent consumption was found likely with the investigation of different anion/cation sources. Although this method is user-friendly and cost-effective, the accuracy and reproducibility of the method mainly depend on the conditions of the electrodes [7,36,66]. Moreover, the optimization of the electrodes may resolve the accuracy and reproducibility problems.

##### pH Metric Methods

In the absence of a chemical laboratory, pH metric methods were invented to determine the FFA value. These methods do not require any specific engines to be applied [67]. The methodology of the pH metric system is the oil samples and the reagent, which are triethanolamine + KNO_3_ + H_2_O + i-PrOH, are added into a beaker and mixed together at ambient temperature. For the calculation of the FFA value, the following equation is applied [67,68]:(2)$$pH=A-logN_{a}$$
where A is the constant value for the given reagent, and N_a_ represents the FFA concentration (mol/L) in the reagent. Nevertheless, this method can be altered to an automated flow pH system to obtain faster results and lower solvent consumption. Gerasimenko et al. [68] designed an automated flow pH system, which is presented in Figure 4, for the total acidity determination of edible oils. In the designed system, the reagent and sample were pumped into a magnetic stirrer and then fed into a metallic coil. After the coil, the emulsion flowed through the pH cell (pHC) with a pH sensor (pHS). Later, the emulsion continued its route to the pH meter (pHM) and the overall acidic percentage was calculated by utilizing the operation processor (OP) [68].

The pH metric methods demonstrate satisfactory results with high accuracy, repeatability, reproducibility, and sensitivity. Moreover, the solvent consumption is 10–15 times lower than the standard titration method, and acidity determination of sample takes approximately 9–10 min to complete [37,67,68]. Thus, this method presents great potential for use as an alternative to the classical titration method to detect overall FFA values.

#### 2.2.3. Spectroscopic Methods

##### Infrared Methods

Among the vibrational spectroscopy methods, infrared spectroscopy gained attention to determine FFA content in oils. Infrared spectroscopy is based on either transmission or reflection measurements (Figure 5).

The most widely used version is Fourier transform infrared (FTIR) spectroscopy. This method has the capability to analyze various chemical compounds through statistical tools within the wavelength range of 400–4000 cm^−1^. FTIR is based on identification of polymeric, chemical, and inorganic materials by utilizing functional group information from functional groups obtained during the scanning of samples with infrared light [40,41,69,70]. An FTIR-based continuous system is capable of analyzing 40 samples per hour according to the study by Al-Alawi et al. [71]. Furthermore, FTIR detectors offer satisfactory sensitivity and precision.

For the qualitative and quantitative determination, near-infrared (NIR) and/or mid-infrared (MIR) spectroscopy may be used regarding the expected wavelength responses, sample size, and applied sample type [72]. During MIR and NIR usage, derivatization methods can be applied to enhance sensitivity and prevent overlapping [41,72]. In some studies, NIR is specified with instrumental advantages, such as robustness and predictability of calibration models [73]. NIR spectroscopy may become preferable owing to the short analysis time, small sample requirement, and developments in hardware design, software, and modeling techniques as a non-destructive method [45,74,75]. The measurable wavelengths of NIR and MIR are listed below:-For NIR spectroscopy, the absorption of electromagnetic radiation at wavenumbers ranging from 12,500 cm^−1^ to 4000 cm^−1^ (~800–2500 nm wavelength) [73].-For MIR spectroscopy, the absorption of electromagnetic radiation at wavenumbers ranging from 4000 cm^−1^ to 857 cm^−1^ [72,76].

Additionally, attenuated total reflectance–Fourier transform infrared spectroscopy (ATR-FTIR) is suitable to be employed at laboratory and industrial scales. It requires small amounts of sample for analysis, such as 0.10 mL, 1 drop of oil with the spectral area of 4000–600 cm^−1^, which is almost same as MIR, and it also offers recording up to 128 scans of each spectrum [77,78,79,80]. Besides, Nascimento et al. [78] propounded that FFA determination in commercial oil samples enforced minimal degradation by ATR-FTIR, better than ^1^H NMR [78], and Triyasmono et al. [81] agreed that the projection pattern of ATR-FTIR was more reliable than ^1^H NMR over FFA determination of crude palm oil [81].

Moreover, the measured infrared regions may be classified, as explained below with respect to existing functional content:4000–2500 cm^−1^: strong contributions from O-H, N-H, C-H, and S-H stretching vibrations are seen in the X-H stretch region,2500–2000 cm^−1^: strong contributions from gas-phase CO (2143 cm^−1^) and linearly adsorbed CO (2000–2200 cm^−1^) are observed in the triple-bond region,2000–1500 cm^−1^: bridge-bonded CO as well as carbonyl groups of adsorbed molecules are seen in the double-bond region,1500–500 cm^−1^: all single bonds between carbon and elements are seen in the fingerprint region, such as nitrogen, oxygen, sulfur, and halogens,200–450 cm^−1^: in the M-X or metal-absorbance region, the metal-carbon, metal-oxygen, and metal-nitrogen stretch frequencies are observed [39,80].

Detection of the metal-carbon bond is hard due to low frequencies (200–450 cm^−1^) [39]. Moreover, it can be specified by usage of atomic force microscope IR (AFM-IR) [82].

In case of sampling of various oils, such as palm oil, deep frying oil, extra virgin olive oil, and fruit oils, in the research, the absorption bands and functional groups were summarized as follows: The cis-C=C bonds at 2924–2855 cm^−1^ were related to the asymmetric and symmetric vibration of the C-H stretching. The C=O stretching of triglyceride ester linkage was visible around 1740 cm^−1^, and C=O band resonance was visible between 1680 and 1780 cm^−1^. The characteristic carbonyl stretching band of FFAs (C=OH) could be seen at 1710 cm^−1^. The absorption of cis-C=C bonds was observed related to the C-H absorption of bending vibrations of CH_2_ and CH_3_ bands from 1654 cm^−1^ to 1460 cm^−1^, and weak absorption might be visible at ~962 cm^−1^. The asymmetric C-O stretching vibrations (ester bonds) were detected at 1236, 1160, and 1117 cm^−1^ [70,77,81,83,84,85]. The specific peak at 725 cm^−1^ was identified as (-CH2)_n-_ bending for sunflower, rapeseed, maize, palm, and coconut oils and their thermal exposure varieties, according to Mitrea et al. [85]. For these oil types, the detected wavelength ranges were from 2924 cm^−1^ to 962 cm^−1^. This approach fits with the literature and may create a reference for FFA detection of various oil examinations. Besides the user-friendly, cost-effective mechanical advantages of NIR, the wavenumber range of NIR-FTIR presents a lack of accuracy over oil sample analysis due to the overlapping problem, while ATR-FTIR and MIR-FTIR offer applicable ranges to detect FFA with high accuracy [74,77,80,86].

Infrared methods present accuracy, high sensitivity, and direct results, whereas the instrument costs and requirements of pretreatment and calibration make this method hard to apply in some cases.

##### Raman Methods

Raman scattering utilizes vibrational transitions to define the molecular structure [38]. Raman spectroscopy utilizes inelastic scattering in contrast to infrared spectroscopy, although all vibrations are visible (Figure 6).

This method is highly suitable for in situ conditions. A disadvantage of this method is the low intensity to detect weak signals [39]. Raman spectroscopy for edible oil sample analysis might be enhanced with FT or chemometrics to enrich the high accuracy of the results and overcome the overlapping issue of the complex forms [88,89]. The significant peaks of Raman spectroscopy are comparable with FTIR.

In the research of El-abassy et al. [38] and Qiu et al. [90], the various types of olive oils were examined by Raman spectroscopy, and comprehensive results are presented in Figure 7. The wavenumber range was applicable between 700 and 3050 cm^−1^. The band at 1265 cm^−1^ could be assigned to δ(=C-H) of cis R-HC=CH-R, and the C-H twist of the -CH_2_ group was at 1300 cm^−1^. δ(C–H) scissoring of -CH_2_ could be seen at 1440, 1650, and 1750 cm^−1^ bands. Furthermore, the symmetric CH_2_, symmetric CH_3_, and cis-RHC=CHR stretches were observed at 2850, 2897, and 3005 cm^−1^, respectively. The three bands, which were at 1008, 1150, and 1525 cm^−1^, corresponded to C-CH_3_ bend, C-C stretch, and C=C stretch, correspondingly [38,90]. The Raman system was noted for operating from 785 nm to predict the fatty acid composition in vegetable oils and to identify waste cooking oils [90].

As seen in Figure 7A, the overlapping issue was observed when using Raman spectroscopy to detect all chemical compounds within the 1750–3050 cm^−1^ wavenumber range. Instead of obtaining clear peaks, more than three peaks were nested together, leading to inadequate compound prediction. In Figure 7B, it was evident that while most of the compounds were the same in different kinds of olive oil based on the positioned peaks, each type exhibited a unique combination of components. Additionally, this method demonstrated a weakness in detecting low-intensity peaks.

##### Nuclear Magnetic Resonance (NMR) Methods

NMR spectroscopy (^1^H, ^13^C, ^31^P) is a great alternative method for FFA determination due to not requiring any pretreatment of the samples. The NMR method is straightforward, and derivatization is not needed. The quantitative results showed that the integral of a signal corresponds to the number of nuclei. For the latter, this method might be applied to the oil and lipid samples that are sensitive to the heat and cannot be analyzed chromatically [8,14,91,92].

High-field (HF) and low-field (LF) NMR can be applied for the acidic property determination of biological samples [91,92]. Although HF NMR presents higher resolution and sensitivity than LF NMR, it has high maintenance costs and requires specialized users [93]. Thus, the most utilized NMR methods are ^1^H, ^13^C, and ^31^P LF NMR for oil sample analysis [14,18].

The ^1^H NMR method is accurate and suitable to classify even minor products, such as phosphorous compounds, of the applied sample, and the spectral range is up to 12 ppm [14,78]. This range was found suitable for analysis of krill oil and VO according to various studies [8,78,94]. Moreover, this method has the advantage of detecting hydro-peroxide and aldehyde amounts in the same spectrum. The usage of 0.6 mL of CDCl_3_/DMSO-d_6_, CCl_4_/DMSO-d_6_, or CS_2_/DMSO-d_6_/CHCl_3_ solvent mixtures in a 20 mg sample might assist in altering separation on the signals. The FFA value of the VOs and fruit oil could be easily detectable in the range of 2.2–2.4 ppm in the form of triplets, and carboxyl content could be measurable from 11 to 12 ppm in the form of a multiplet [8,95,96], whereas a strong signal could cause overlapping in this method [8,14]. For the acidic property determination of the VO sample, the ^1^H NMR method presents a spectral width of 24 ppm, number of scans of 32, relaxation delay of 1 s, and acquisition time of 5.45 s [97]. Schripsema et al. [98] investigated aqueous extracts of butter and margarine samples at 25 °C, with a spectral width of 10 ppm and a relaxation delay of 4 s, and the results were precise and comparable with GC and HPLC [98]. The levels of unsaturation and stability of the oil matter to obtain accurate results. In case of overlapping, this method might be coupled with partial least squares (PLS), GC, or Raman methods [8].

The ^13^C NMR method has a large spectral width (>170 ppm), covering and obtaining accurate results with a wide range of product determination [14,78]. Even though various studies showed that this method required a longer relaxation time, such as 20–25 s [14], the ^13^C NMR method was applied by Monakhova et al. [94], whose study presented high accuracy for animal origin oil analysis with 256 scans, a 256k time domain, a relaxation time of 2 s, and an acquisition time of 3.36 s [94]. Moreover, the study of Nascimento et al. [78] offered reliable and effective results for VO analysis with a spectral width of 200 ppm, acquisition time of 2.17 s, and relaxation delay of 0.98 s via the ^13^C NMR method [78]. Also, the relaxation reagents can be added to obtain a shorter experimental time, such as chromium (III) acetylacetonate (Cr(acac)_3_) [14]. While esterified fatty acids could be observed over 173–175 ppm, FFAs could be detected over 176.5–181 ppm with this method for krill oil analysis [94,99]. This FFA detection range was 176–18 ppm for the VO sample and 180–182.78 ppm for the UCO sample according to di Pietro et al. [14].

The ^31^P NMR method is accurate, similar to other NMR methods. Also, this method is effective for edible, waste, and recycled oil analysis [14]. For the analysis of VO, the ^31^P NMR method is run with a spectral width of 100 ppm, number of scans of 512, relaxation delay of 10 s, and acquisition time of 6.50 s [97]. The FFA values of VOs could be visible at 134–136.5 ppm, glycerol contents at 146.5–148.5 ppm [14,100], and phospholipids could be visible at 0.4–(−1) ppm [99,101]. Moreover, this method presents advantages, such as faster resolution, sensitivity, etc. The chemical shift range covers more than 700 ppm. Furthermore, it can be utilized to analyze solid and semisolid samples, as well as the liquid samples [100].

NMR and infrared methods can be alternatives to each other due to being non-destructive and capable of rapid detection and high efficiency. For the accuracy and reliability of the results, the type of applied sample should be considered since each NMR method has a different detection range. In Table 1, a comparison of these two methods is presented.

#### 2.2.4. Chromatographic Methods

Chromatographic techniques are easy to use for analytical measurement. The characterization of the chromatographic process includes evaluation of efficiency, sample capacity, etc. Despite the fact that there are many chromatographic techniques, the most common ones are gas chromatography (GC) and high-performance liquid chromatography (HPLC) [25].

##### Gas Chromatography (GC)

Gas chromatography (GC) is a spectroscopic method and a powerful technique for the analysis of volatile fatty acids. GC provides quantitative and effective results, although it suffers from long analysis times [49]. Gas chromatography–mass spectrometry (GC-MS), flame ionization detector (GC-FID), and tandem mass spectrometry (GC-MS/MS) are common spectroscopic methods used for the analysis of fatty acids in complex biological matrices, such as oils (edible and harvest), animal fats, fish oils, dairy products, etc. This methodology does not require any sample pretreatment, making it highly convenient for straightforward application [84,102]. For the FFA determination, the temperature range is foreseen from 40 °C to 240 °C, and the exact range depends on the applied sample [102].

GC-MS may detect more structural information, and present accurate databases for identification of fatty acids with higher efficiency and better selectivity than GC-FID [84]. However, lipid species require a high boiling point. Thus, labor preparation is necessary for sample analysis in GC-MS to form and detect FFA of the triglycerides (TAGs), because all compounds should be volatile in the GC-MS method, and it can cause transformation of triglycerides (TAGs) to their relative FFAs due to utilizing a high column temperature [48]. Moreover, GC-MS is capable of detecting FFAs when they are neutralized by saponification, separation, and esterification [16]. Additionally, GC-MS/MS was found beneficial to define the fatty acid profile of complex biological samples with strong structural identification capability by Chen et al. [103], Herrmann et al. [104], and others thanks to the dual determination system. MS/MS application has not been confirmed for FFA detection of VO whose content presents high amounts of unsaturated C_18_, and has been revealed as a suitable technique for detailed FFA determination of VO by Beneito-Cambra et al. [105]. This claim has paved the way of further investigations.

The utilized reagents have as much importance as the methodology to obtain proper identification in GC-MS. Thus, Nzekoue et al. [106] presented valuable research from this perspective and investigated salting-out agents, such as NaCl, Na_2_SO_4_, NaH_2_PO_4_, and Na_2_CO_3_, to detect FFA (Figure 8). The solubility and ionic strength of NaH_2_PO_4_ were higher than other agents in the extraction of samples. NaCl, Na_2_SO_4_, and Na_2_CO_3_ competed in the analysis by decreasing their solubility. This developed method was useful to increase efficiency and detect short/mid-FFA chains, especially in lipids, e.g., cheese [106].

Gas chromatography–combustion–isotope ratio–mass spectrometry (GC-C-IR-MS), gas chromatography–ion mobility spectrometry (GC-IMS), and flash gas chromatography–electronic nose (FGC-Enose) are altered gas chromatography methods used to improve sensitivity in compound prediction. In IMS, the ion–ion interaction is taken into account as a working principle. In the latter, FFC-Enose couples with two parallel short columns, each coupled to a flame ionization detector (FID). For GC analysis, the applied temperature range is around 200–300 °C in general. However, GC-C-IR-MS utilizes up to 900 °C due to the combustion methodology [24,107]. As an example, GC-IMS and FGC-Enose chromatograms are employed to present sensitive and accurate detection for extra virgin olive oil (EVOO) and soft deodorized oil in Figure 9.

GC-IMS utilizes 3D plots by employing various test parameters to identify the optimal methodology for detecting clear peaks and determining the content of EVOO. These plots enable a comprehensive visualization of the data, helping researchers to achieve accurate peak detection, ultimately improving the analysis of EVOO composition (Figure 9A,B). Likewise, FGC-Enose presented clear peaks with high sensitivity to determine FFA content in Figure 9C,D. Moreover, it is obvious that deodorization of oil led to removal of FFAs from the oil composition in Figure 9B,D.

GC analyses are feasible, accurate, reproducible, and sensitive. Nevertheless, the utilized samples have to be volatile to detect FFAs, and this can affect the components’ composition during detection owing to heat application.

##### High-Performance Liquid Chromatography (HPLC)

High-performance liquid chromatography (HPLC) is a kind of liquid chromatography. This technique does not require high temperatures, and semi/nonvolatile samples may be utilized at ambient temperature [25,108]. Furthermore, HPLC demonstrates the benefit of being capable of detecting a wide spectrum of lipids, such as TAG, FFA, phospholipid (PL), lysophospholipid (LPL), etc., while alternative methods, such as GC and NMR, exhibit limitations in this regard [48,109,110].

HPLC converts FFAs to a large number of different derivatives by utilizing various labeling reagents, such as 2-(11H-benzo[a]carbazol-11-yl)-ethyl-4-methylbenzenesulphonate (BCETS) and acridone-9-ethyl-p-toluenesulfonate (AETS). These reagents react with FFAs in the selected co-solvent to obtain high yields of esters in the presence of the chosen catalyst (K_2_CO_3_, K_2_C_2_O_4_, Na_2_CO_3_, P_2_O_5_, etc.) [111,112]. The derivatization of FFAs presents some advantages, such as high sensitivity, detection of a wide range of products, regardless of their polarity and chain length, and prevention of overlapped peaks. Furthermore, it is easy to calibrate, eco-friendly, cost-effective, and has low toxicity [48,111,113].

The HPLC system is mostly coupled with a fluorescence spectrometer to obtain higher sensitivity [65,111]. Li et al. [111] developed the BCETS reagent for determination of FAs in edible oils via HPLC-fluorescence detection. In the study, BCETS was dissolved in acetonitrile, dried with P_2_O_5_, and mixed with FAs to convert FAs into their derivatives (Figure 10). This solution was diluted with ACN/DMF (1:1, *v/v*). This diluted mixture was injected into a chromatogram. The sharp peak responses were obtained via an approximately 70 min retention time with a 30 °C column temperature, were clear, and the R^2^ values were >0.9994 in the study. Saturated and unsaturated FFAs were detectable using this methodology [111].

Moreover, the HPLC system can be employed with various alternative systems, most commonly evaporative light scattering detector (ELSD), charged aerosol detector (CAD), diode array detector (DAD), UV, MS, and Ag^+^, for FFA detection of biological samples. Previous studies have demonstrated that HPLC-ELSD yields inconsistent data owing to the non-linearity of standard curves and high limit of detection of FFA, which occurs due to the sensitivity of the flow rate and temperature [110,114,115]. In the principle of CAD, the number of signals is generated by utilizing an electrometer, which charges particles, and CAD is deliberated for being more sensitive than ELSD. Additionally, HPLC-CAD was described as an effective method for lipid content determination, with a 20 µL injection volume at 65 °C column temperature and a 50 °C CAD evaporation temperature, by Kim et al. [110]. In the FFA determination of canola and red palm oils by HPLC-ELSD, adequate accuracy was unlikely to be observed due to the non-linear standard curve, according to the study of Choi et al. [114].

Likewise, Zhao et al. [116] mentioned that UV and MS share the same weakness as ELSD, and the MS instrument is also more expensive than CAD for applying sample analysis. Moreover, the classical GC method, utilized as a reference methodology, overestimated oleic acid %, and HPLC-CAD was helpful to detect more FA components in their research [116]. The retention time of HPLC-CAD is approximately 15–30 min, which represents a relatively fast analysis [48,110,115].

The HPLC-DAD method was found beneficial for determination of phenolic compounds, whose accumulation in the oil might be problematic for human health, such as tyrosol, along with the official method of the International Olive Oil Council (2009) by Ozbek et al. [117]. Moreover, fatty acid profiles of seed oils were investigated with HPLC-DAD for evaluation of the quality of the oils as bioactive-rich compounds (tocopherols, phytosterols, and others) by Ozbek et al. [117] and Lyashenko et al. [118]. Yuenyong et al. [119] experimented with the effectiveness of HPLC-DAD via an analysis of 50 different plant oils (0.5 g), which were diluted with dichloromethane until reaching the final volume of 1.00 mL. The solution was filtered (0.45 μm) before HPLC injection. The utilized mobile phases were methanol and water, operating in a gradient mode. The FFA determination results were obtained within 30 min, and they were precise and comparable with GC-MS results even for the application of pseudo-cereal oils, such as cauliflower, leafy green seed oil, and others [119].

One of the oldest ways to determine the fatty acid profile of lipids is Ag^+^-HPLC, whose column is silver-impregnated [120]. The silver ions form polar complexes with unsaturated centers of fatty acids, reversibly, though the residual silanol groups of the column create an interaction with carboxylic acid groups of FAs [121]. This method was described as effective, especially in the determination of conjugated linoleic acid (CLA; which is a mixture of linoleic acid isomers, especially cis-9,trans-11 C18:2 (c-9,t-11 C18:2)), in biological samples, such as food products from ruminant origin and some plant products, containing a conjugated double bond and valuable ingredients in dietary supplements without derivatization, by Ostrowska et al. [120], Luna et al. [122], Yurawecz et al. [123], and Białek et al. [124]. Moreover, CLA isomers displayed beneficial effects in arranging the fatty acid composition of blood, antidiabetic, and anticancerogenic properties with limited daily usage (2–3 g per day) due to the ability of nutrient adsorption in humans [124]. All samples required methylation (FAME production) to detect isomers. The study of Luna et al. proved that utilization of an acidic catalyst (i.e., BF_3_ or HCl) and mild conditions are essential for accurate determination [122]. Czauderna et al. performed various studies on the determination of CLA by Ag^+^-HPLC, and one of their studies pointed out that the combination of mild saponification with base- and acid-catalyzed methylation at 40 °C for 1 h prevented further isomerization during sample analysis [125]. Moreover, the hydrolysis method might be required for the accurate FFA analysis of animal fats, VO, and lipids regarding their complex structure [122,126]. Hydrolysis parameters depend on the tested material and utilized catalyst. Nevertheless, the previous studies showed that hydrolysis application as a preparation step of HPLC was helpful, with a specific temperature (60–90 °C) and duration time (30 min–2 h), for accurate determination of poly-unsaturated fatty acids, glycerides, and others, which were adsorbed in the structure [121,122,126,127]. Additionally, Czauderna et al. [121] combined Ag^+^-HPLC with DAD, which operated with a UV range from 195 to 400 nm with a 1.2 nm spectral resolution and 1 spectrum per s measurement frequency, to define CLA isomers. They prepared n-hexane/acetic acid/acetonitrile (98.4:1.6:0.0125 *v/v/v*) solution as a mobile phase and equilibrated the column for 40 min before sample injection. Then, biological samples were prepared via the hydrolysis method with 2 M NaOH (3.5–4.0 mL) at 80–85 °C for 30–35 min, followed by acidification with 4 M HCl to pH ~2, extraction of FFAs with 3.5 mL of dichloromethane, drying with Na_2_SO_4_ (50–100 mg), and solvent removal under Ar. They confirmed that this altered system had great agreement with separation of geometric and positional isomer mixtures, e.g., trans-trans, cis-trans, trans-cis, and cis-cis configurations, in a narrow UV range (228–240 nm). CLA isomers were particularly detected at 234 nm [121]. In another study, by Czauderna et al. [128], the original internal standards (IS) were used in an analysis of CLA and/or conjugated trienes isomers by Ag-HPLC-DAD-UV. Biological samples were prepared with 1 mL of 1M KOH in methanol, 1 mL of 2M KOH in water, and 25 μL of IS (6 mg of sorbic acid (c2c4C6:2) in 15 mL of chloroform), followed by flushing with Ar stream for 1 min and left overnight at room temperature for hydrolysis. Then, 1.5 mL of water was added, and the solution was acidified with 6 M HCl to pH ≈ 2 to produce FAME. For the latter, this sample was extracted with 3.5 mL of dichloromethane, dried with Na_2_SO_4_ (50–100 mg), and connected to an Ar stream for solvent removal. As a result, CLA isomers were detected at 234 nm, as in previous research [121], and IS was used to assist in the removal of impurities from mobile phase and in altering baseline noise for better detection, which was observed at 259 nm [128].

These coupling improvements were helpful to improve the sensitivity and accuracy of HPLC. Furthermore, the HPLC methodology has easy application and sample preparation procedures.

#### 2.2.5. Thermometric Titration Method

The thermometric titration (TT) method is a kind of titrimetric method. In automated titration systems, KOH in isopropanol NaOH solution or n-butylamine in toluene/benzene may be utilized as titrants, and data are saved by a detector. Hence, dissolved n-butylamine in toluene/benzene was found suitable to decide the total acid value of catalysts, by Ilnicka et al. [31], while isopropanol and NaOH solution were generally utilized on edible oil samples [30,31,129].

In this method, the heat is provided to the system and the heat changes are measured up to the end point during auto-titration. The temperature range is generally 18–25 °C [58,130]. Moreover, it is suitable to combine with the IR method to obtain more precise results. Alessio et al. [30] studied TT coupled with the IR method to detect the acid value of the samples by using neutralization, complexation, and redox reactions, respectively, and they demonstrated quantitative results, which were comparable with conventional methods [30].

This method may be applied to colored oils, whose acidity is high, and it is more preferable over classical titration method thanks to having advantages such as high analysis speed, simplicity, less solvent consumption by sensor usage, and low cost [29], whereas it presents an overall FFA value, as in the classical titration method.

## 3. Alternative Methods for FFA Determination

Industrial and classical methods are explained in previous sections, with their advantages and disadvantages. Lately, researchers have paid attention to altering the existing FFA determination methodologies of VOs and UCO while preserving the inherent advantages due to environmental concerns.

Advantages of utilizing laboratory-based hyperspectral imaging (HSI) have been researched to detect the peroxide value (PV) and FFA. The researchers combined HSI with NIR spectroscopy and tested different oil samples in a 400–1000 nm spectral range. Even though they presented promising results, further research ought to consider the calibration [131].

Various fast kits were invented for FFA determination, although many of them remained as hybrid methods. Based on the physical and chemical parameters of oil, some of the fast kits are 3M TM Oil Quality Test, Strips, FASafe, and CDR FoodLab^R^, which were produced to save time in the determination of oil quality and FFA (Table 2) [26,132].

The principles of the 3M TM Test strips, FASafe, and CDR FoodLabR are schematized in Figure 11. The 3M TM Test strips have two ranges, a lower range of 0–2.5% FFA and a higher range of 0–7% FFA, which indicate reliable fast information about the quality of the oil by following the color change from blue to yellow on the strips (Figure 11A) [133]. FASafe is a calorimetric method. Small amounts of sample were treated with the selected reagent at 37–44 °C for 10 min to determine the FFA value (Figure 11B) [132]. The principle of CDR FoodLab^R^ is based on measuring the acid value on the digital screen by using an oil-resultant blend (Figure 11C).

The FASafe method is accurate, fast, reproducible, and more expensive than AOCS owing to utilizing a solvent, ethanol, and one duplicate trial costs USD 2.86 [26]. The repeatability and reproducibility of the FASafe method with the comparison of the AOCS titration method, as a reference, is presented in Figure 12. AOCS titration and FASafe spectrophotometric methods present similar distributions for FFA detection. Thus, this FASafe method is promising and reliable as an alternative method.

Ultra-high-performance liquid chromatography–quadrupole time of flight–mass spectrometry (UPLC-Q/TOF-MS) may also be utilized for fast determination of FFA. Gao et al. [134] studied FFA determination of herbal oil with the following experimental conditions: scan range of 50–1000 *m/z*, 0.2 s scan time for each function, and temperature up to 500 °C. This method was found to be fast and accurate and it did not require pretreatment of samples, although calibration was necessary. All experimental results were comparable with GC-FID results [134].

Recently, fluorescence and UV spectroscopies have gained interest for FFA determination since a limited number of samples can be detected by the FTIR method (Figure 13). The spectra of olive oil samples might be between 200 and 800 nm in UV spectroscopy and 300–800 nm in fluorescence spectroscopy, while this range is 4000–450 cm^−1^ for FTIR [135,136]. The primary and secondary oxidation products might be determined by UV spectroscopy, whose data utilizes the matrix form for determination and prediction [17,135].

A fluorescence spectroscopy system does not require any expensive maintenance. The main parts of the fluorescence spectroscopy system are exhibited in Figure 14. In the principal system, the tested sample emits light from the LED sources, and the data are measured and recorded by a camera.

Fluorescence and UV spectroscopies are effective for oil type separation, such as EVOO and VOO. However, these methods are weak for a wide range of compounds’ determination, although they have been found to be fast, accurate, reproducible, and non-destructive [111,136].

## 4. Comparison of Different FFA Analysis Methods

The utilized FFA analysis methods of biological materials were described in the previous sections. Among all FFA analysis methods presented in Figure 2, the classical titration method is well known and is the oldest method for the determination of the overall FFA value. It presents high solvent consumption, large sample requirements, and a lack of accuracy. Thus, these limitations led to motivation for its improvement. Instrumental methods were developed as an alternative to the classical titration method. Since this method is classified into different methodologies, the fundamental properties are presented in Table 3 to provide a clear and concise comparison of the methods discussed. All these methods are applicable in FFA determination of oil samples, as targeted.

The overall value of FFAs might be defined by colorimetric, thermometric titration, and electrochemical methods. This leads to inadequate FFA determination, as with the classical titration method. Solvent consumption is moderated, and the solvent requirement is still high, which makes them harmful for the environment. FIA and voltametric methods share a potential of coupling with the HPLC method, making them suitable and accurate for various applications [61,65]. Nevertheless, voltametric methods are highly effective for determination of heavy and toxic metals [63].

Spectroscopic and chromatographic methods provide detailed information on FFA content, which is crucial for understanding the quality of oils. These methods require only a minimal amount of solvent and sample, offering high accuracy, repeatability, reproducibility, and sensitivity for the determination of FFA (Table 3). Although the Raman method is feasible, its accuracy in FFA determination is limited due to overlapping peaks, which might be improved by coupling it with FT [39,88]. For unsaturated FFA determination, NMR and infrared methods are among the most accurate ones, as detailed in Table 1, and might also be used for solid sample analysis, such as butter [98], cheese [137,138], and meat [139] products. Besides, HPLC is a versatile alternative for determining fatty acid profiles of various sample types, as it does not require high heat application. It has flexibility to be employed with different methods, such as florescence, FTIR, and others, to enhance the accuracy and sensitivity. Ag^+^-HPLC is effective for lipids and unsaturated FFA determination, with slightly higher solvent and sample requirements than NMR and infrared methods. HPLC-CAD is advantageous for FFA analysis of oil samples, such as VO, while HPLC-DAD is preferable for analyzing phenolic compounds. Despite the requirement for volatile samples due to heat application, GC remains a well-known method. GC-MS/MS is particularly beneficial for analyzing the fatty acid profile of biological samples, such as cereal oils, due to its dual detectors, which prevent overestimation of weak signals [103,104].

## 5. Conclusions

The determination of free fatty acids (FFAs) is critical for assessing the quality of oils used in various industries. This review highlighted the strengths and weaknesses of different FFA determination methods, including classical titration, chromatographic techniques, and spectroscopic methods.

The classical titration method is highly toxic due to its substantial solvent requirements. Even though colorimetric, electrochemical, and thermometric methods have less solvent consumption than the classical titration method, except copper soap and EC methods, they still rely on solvents, thereby lacking an eco-friendly approach.

Colorimetric, electrochemical, and thermometric methods offer faster results than spectroscopic methods. Trained personal, high maintenance costs, and calibration are not required for their application. The main weaknesses of these methods are high solvent usage, low frequency, stability obstacles, overlapping of the results, accuracy, and reproducibility problems, especially for methods such as copper soap, FIA, voltametric, and EC, which pose significant drawbacks. Regarding the latter, they only provide the total FFA value instead of detailed chemical compositions.

On the other hand, spectroscopic (infrared, Raman, NMR, and chromatographic) methods provide rapid and non-destructive analysis, making them suitable for a wide range of applications, such as analyzing oil and lipid samples. The main disadvantages are the need for sophisticated equipment and expertise. Nevertheless, infrared, Raman, and chromatographic methods require sample preparation, unlike NMR. Unless the instrumental cost is an obstacle for usage of NMR, it stands out as a promising method due to its excellent analysis results and minimal weaknesses. Raman spectroscopy offers promising results if the issues of overlapping and low-intensity peak determination are overcome. Especially, NIR spectroscopy has gained attention for being non-destructive, despite the color dependency of the applied sample. Among all chromatographic methods, HPLC-DAD, Ag^+^-HPLC, Ag^+^-HPLC-DAD-UV, and GC-MS/MS offer high sensitivity and detailed analysis for phenolic compounds and biological samples (i.e., cereal oils), while UPLC, HPLC-CAD, and GC-MS methods are more precise for edible oil analysis.

Fast FFA determination kits, such as the 3M TM Oil Quality Test, Strips, FASafe, and CDR FoodLab^R^, are offered as alternative rapid test methods to detect the FFA value. Inherently, these fast determination methods suffer from the same limitations as other methodologies, except for spectroscopic methods. Thus, fluorescence and UV spectroscopies have gained attention as alternative approaches for rapid determination of FFAs. Although they are not capable of detecting a wide range compounds on their own, they can be effectively combined with other methods, such as FTIR and HPLC, to enhance the sensitivity in analyzing oil and lipid samples.

A suitable FFA determination method is required to allow accuracy, high sensitivity, repeatability, and reproducibility, and it is determined by use of an eco- and user-friendly detection system, applied samples, purpose of the work, and requirement of trained workers. This study suggested that combining methods, such as FIA-HPLC, fluorescence-FTIR, or Raman-FT, or coupling with enhanced detectors using improved solvent types, can overcome the current weaknesses. Future research should focus on developing eco-friendly and cost-effective methodologies that maintain high accuracy and reproducibility. By selecting the appropriate method based on sample type and analysis requirements, researchers can ensure precise and reliable FFA determination.

## Figures and Tables

**Figure 1 foods-13-01891-f001:**
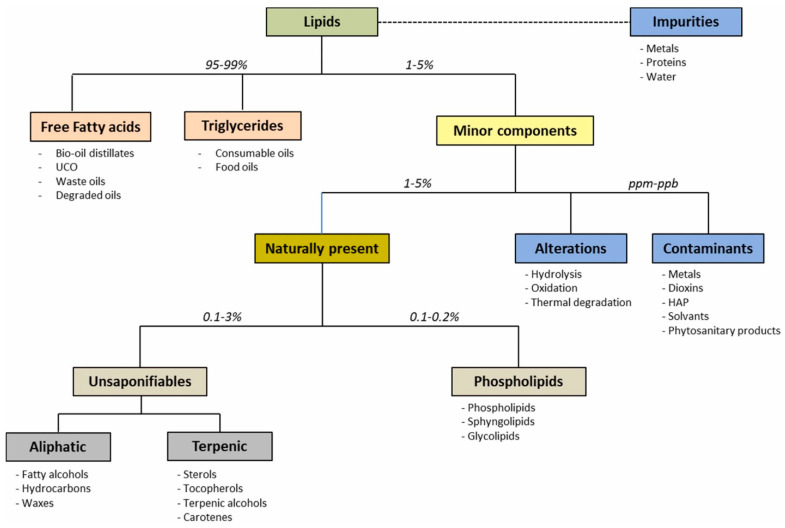
General composition of vegetable oils and animal fats for biofuel production [12].

**Figure 2 foods-13-01891-f002:**
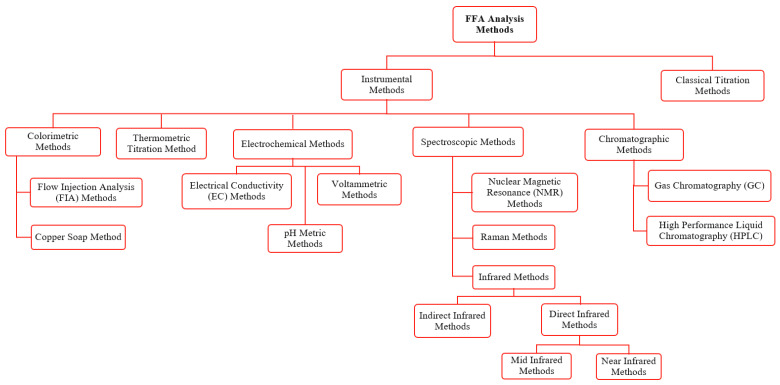
Scheme of various methods for FFA analysis.

**Figure 3 foods-13-01891-f003:**
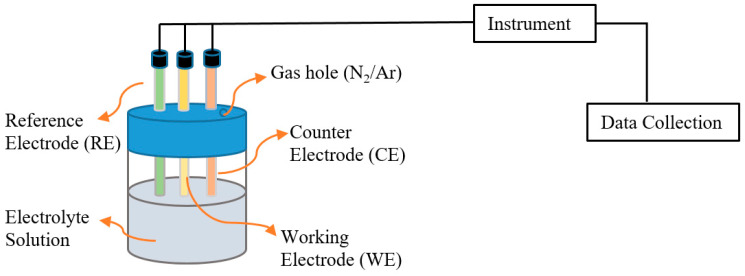
Scheme of a typical three-electrode cell in the voltammetry technique.

**Figure 4 foods-13-01891-f004:**
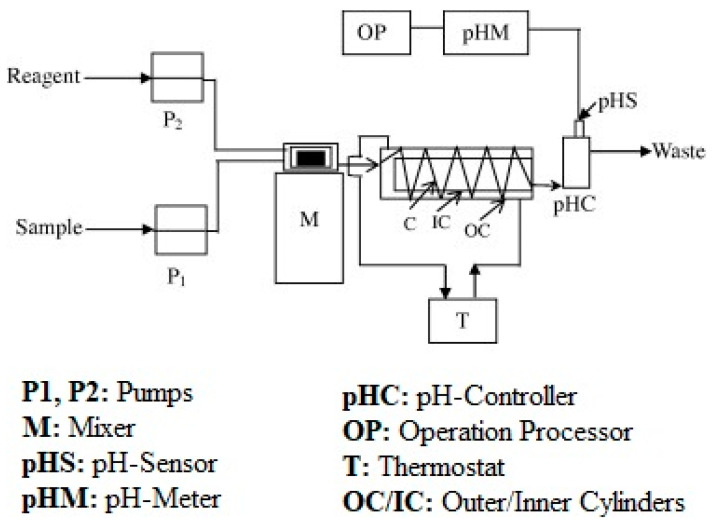
Schematic diagram of the automated flow pH system for the FFA determination in edible oils [68].

**Figure 5 foods-13-01891-f005:**
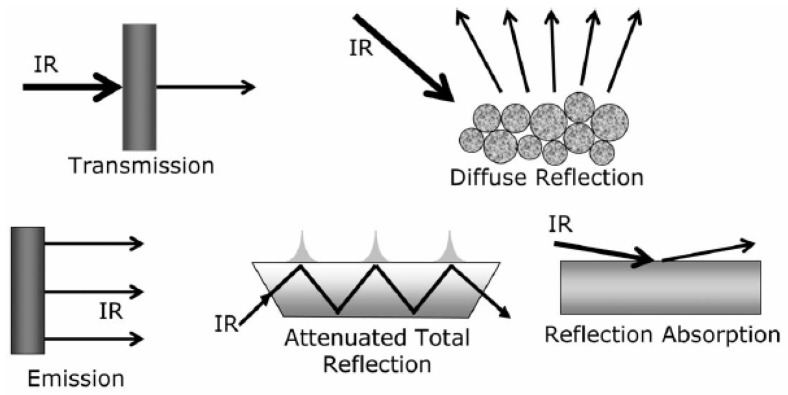
Different ways to perform vibrational spectroscopy [39].

**Figure 6 foods-13-01891-f006:**
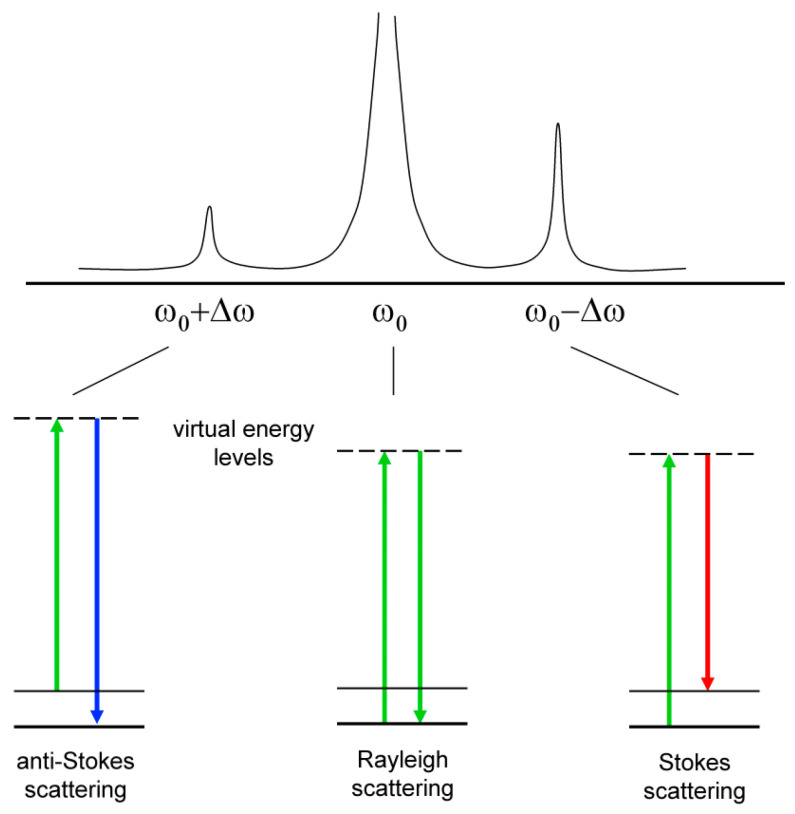
Monochromatic light of frequency numbers is scattered by a sample, either without losing energy (Rayleigh band) or inelastically, in which a vibration is excited (Stokes band), or a vibrationally excited mode in the sample is de-excited (anti-Stokes band) [87].

**Figure 7 foods-13-01891-f007:**
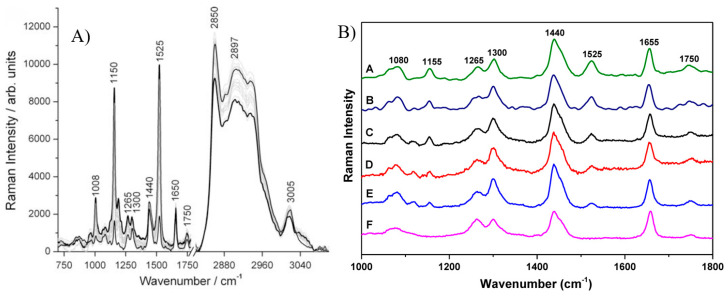
Raman spectra of (**A**) 19 different virgin olive oil (VOO) brands (the grey specs demonstrate the variation width of the Raman data) [38], and (**B**) olive oil samples A, B, C, D, E, and F of different origin [90].

**Figure 8 foods-13-01891-f008:**
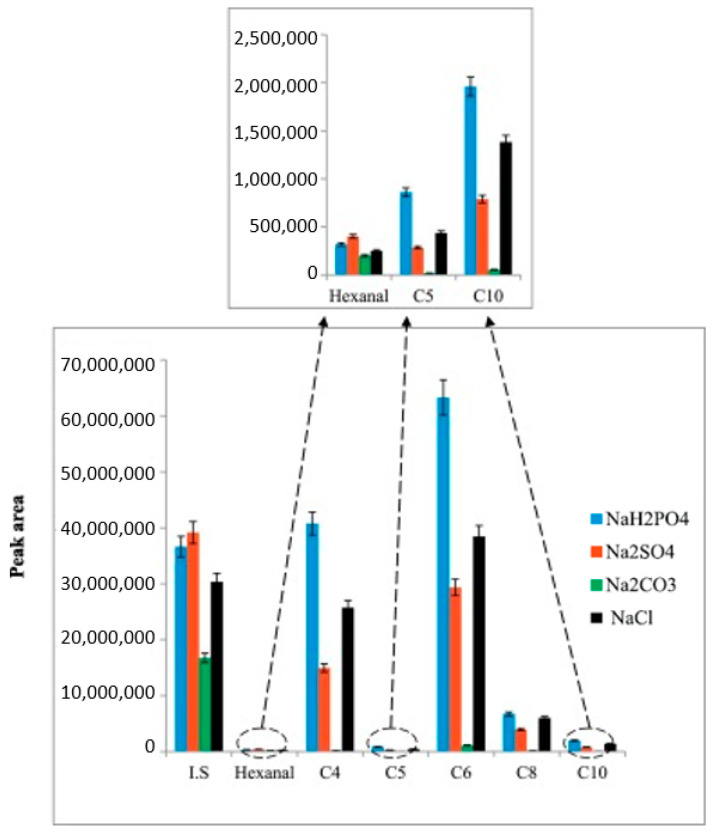
Effect of different salting-out agents in the release of volatile compounds into the headspace. I·S (internal standard): 2-methylpentanal, C4 (butanoic acid), C5 (isovaleric acid), C6 (hexanoic acid), C8 (octanoic acid), C10 (decanoic acid), NaCl (sodium chloride), Na_2_SO_4_ (sodium sulfate), NaH_2_PO_4_ (monobasic sodium phosphate), and Na_2_CO_3_ (sodium carbonate) [106].

**Figure 9 foods-13-01891-f009:**
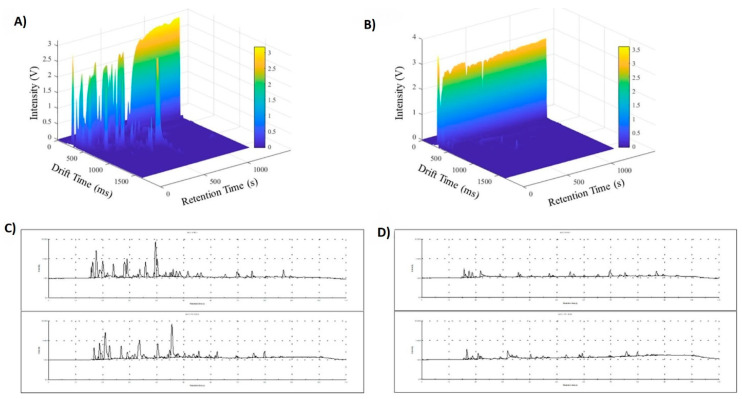
GC-IMS 3D topographic plot of EVOO (**A**) and soft deodorized oil (**B**). FGC-Enose chromatograms of EVOO (**C**) and soft deodorized oil (**D**) [24].

**Figure 10 foods-13-01891-f010:**
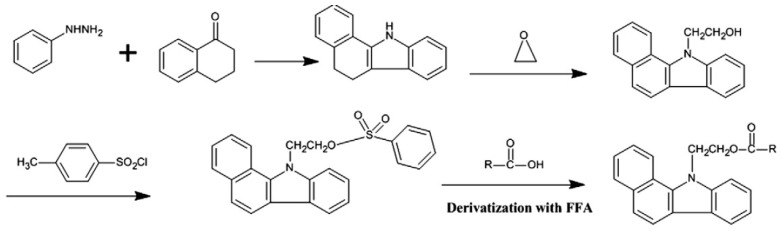
Scheme of BCETS synthetization and derivatization route of FAs [111].

**Figure 11 foods-13-01891-f011:**
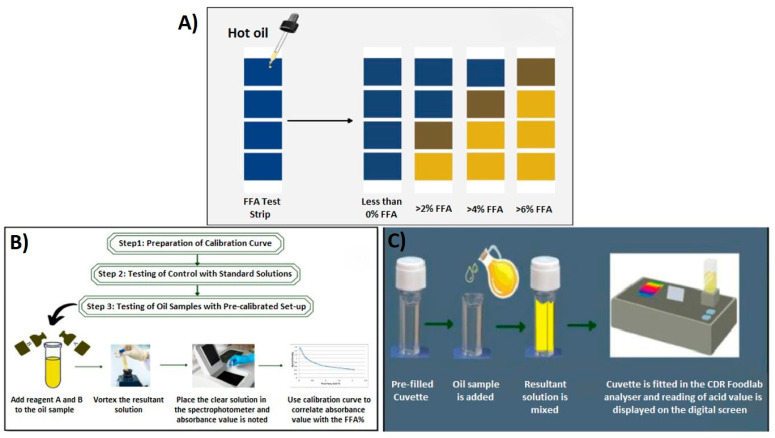
Principles of (**A**) 3M TM Test strips, (**B**) FASafe, (**C**) CDR FoodLab^R^ [132].

**Figure 12 foods-13-01891-f012:**
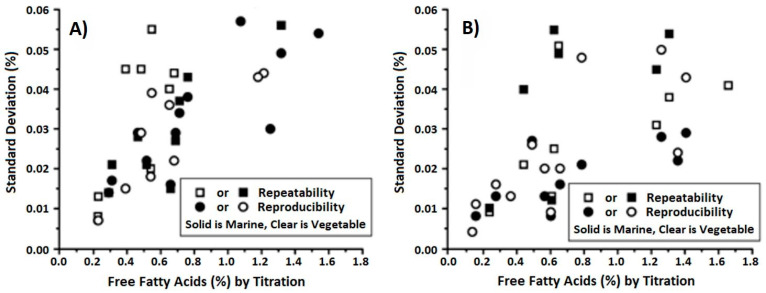
(**A**) AOCS titration method and (**B**) the FASafe spectrophotometric method for determination of FFA levels in marine and vegetable oils [26].

**Figure 13 foods-13-01891-f013:**
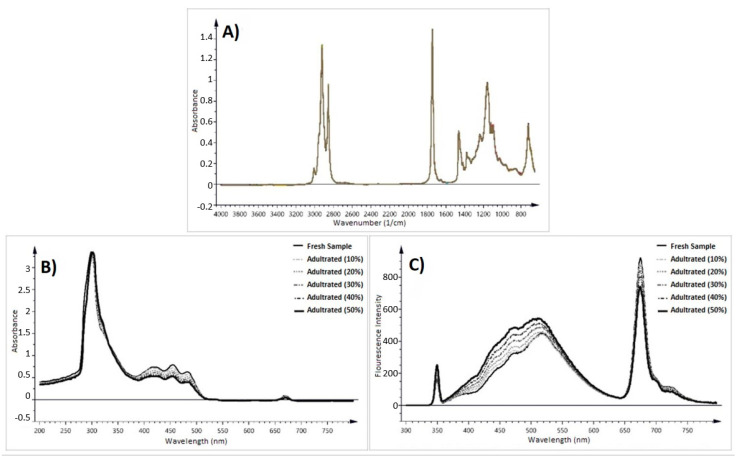
(**A**) FTIR, (**B**) UV-vis, and (**C**) fluorescence spectra of the olive oil samples [135].

**Figure 14 foods-13-01891-f014:**
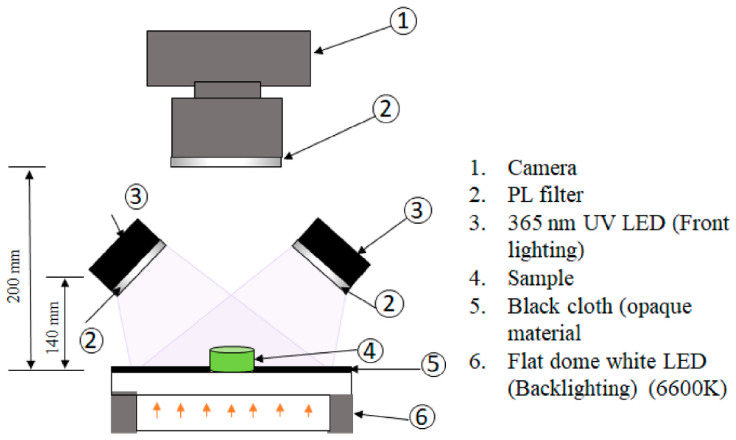
An example of a fluorescence spectroscopy system [136].

**Table 1 foods-13-01891-t001:** Comparison of LF-NMR and infrared spectroscopy [42].

LF-NMR	Infrared
Easy to apply.	Trained personnel needed to apply.
Non-destructive method and sample preparation is not necessary.	Non-destructive method and sample preparation is required.
NMR method is surface-independent.	Since the surface of the samples is utilized for analysis, it is a surface-dependent method.
Color-independent.	Color of the sample can affect the IR spectra, so different calibration curves are necessary regarding sample color and size.
NMR calibration curves require 3–5 reference samples.	Minimum of 10 reference samples are necessary for creating a calibration curve.
For calibration, a simple linear regression fit can be easily applied.	For calibration, different methodologies can be applied, such as derivatization methods, the partial least squares (PLS) regression method, etc.
High reproducibility of analysis.	Reproducibility of analysis is good.
The quantitative results show that the integral of a signal corresponds to the number of nuclei by utilizing the whole sample.	The surface of the sample is utilized for the analysis.

**Table 2 foods-13-01891-t002:** The commonly utilized test kits and producers for FFA determination.

Test Kits	Producers	Limitations	Ref.
3M TM Oil Quality Test	3M Commercial Solutions, St. Paul, MN, USA	This test gives an idea about the range of FFA, instead of presenting detailed compound information.	[132]
Strips	Divisions, Newport, KY, USA	This test gives an idea about the range of FFA, instead of presenting detailed compound information.	[132]
FASafe	MP Biomedicals, Irvine, CA, USA	Laboratory environment, chemical usage, and trained personnel needed.	[17,26]
CDR FoodLab^R^	CDR Mediared Co., Ltd., Firenze, Italy	Validity and reliability of the results have not been proven.	[132]

**Table 3 foods-13-01891-t003:** Comparison of instrumental methods.

Instrumental Methods		Solvent Requirement	Data Variability	Strengths	Weaknesses
**Colorimetric** **Methods**	**Copper Soap Method**	Relatively High	Overall value of FFAs	Simplicity, low cost, reproducibility	Limited determination on dark-colored samples Inadequate determination of FFA
	**FIA Method**	Moderated		Sensitivity, accuracy	Stability Problem Inadequate determination of FFA
**Thermometric Titration Method**		Moderated	Overall value of FFAs	Simplicity, low cost, reproducibility	Inadequate determination of FFA
**Electrochemical** **Methods**	**Voltametric** **Method**	Moderated	Overall value of FFAs	Low cost	Overlapping and reproducibility problems Not applicable for viscous sample Inadequate determination of FFA
	**EC Method**			Low cost	Accuracy and reproducibility of the method mainly depend on the conditions of the electrodes Inadequate determination of FFA
	**pH Metric Method**			Low cost, accuracy, repeatability, reproducibility, sensitivity	Inadequate determination of FFA
**Spectroscopic** **Methods**	**Infrared Method**	Low	In-depth understanding of FFAs	Non-destructive, accuracy, repeatability, reproducibility, sensitivity	Sample preparation is required Calibration could be required in colored samples
	**Raman Method**				Overlapping Limitation for detecting low-intensity peaks
	**NMR Method**				Expensive Trained person needed
**Chromatographic** **Methods**	**GC**	Low	In-depth understanding of FFAs	Feasible, accuracy, repeatability, reproducibility sensitivity	Sample preparation is required Heat application might affect the component composition
	**HPLC**				Sample preparation is required Coupling with other methods could be required for sensitivity and accuracy

## Data Availability

No new data were created or analyzed in this study. Data sharing is not applicable to this article.

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
