# Peer review of "A Comprehensive Study for Determination of Free Fatty Acids in Selected Biological Materials: A Review"

_foods, 2024, doi:10.3390/foods13121891_

Round 1
Reviewer 1 Report
Comments and Suggestions for Authors
Dear AUTHORS,
In my opinion, the review work would be scientifically valuable if it described the determination of fatty acids in biological materials using instrumental analytical methods, especially chromatographic methods: Ag-liqiud-chromatpgraphy, UPLC/HPLC-DAD/FLU/MS or GC with MS and MS/MS.
Lines 120-121, 223-236: FFAs profiles/concentrations were analysed in oil animal fat samples in alkaline solutions (i.e., after saponification !) Therefore: the title of the submitted manuscript misleads readers.
Your review on the determination of fatty acids using methods that also specify the procedures for hydrolysis of samples of animal fats or vegetable or fish oils would be more scientifically valuable. Conditions of hydrolysis methods (like temperature, duration) have a significant impact on the profile of mono- and poly-unsaturated fatty acids (especially conjugated fatty acids, e.g. in milk fats or pomegranate seed oils).
Therefore, I suggest the more interesting scientific scope and title of your manuscript: "A comprehensive study for determination of fatty acids in selected biologial materials: A Review" .
Author Response
Response to reviewers
Manuscript Number: foods-3039246
Title: A Comprehensive Study for Determination of Free Fatty Acids in Selected Biological Materials: A Review
Authors: Beyza Uçar 1,*, Zahra Gholami 1, Kateřina Svobodová 1,2,*, Ivana Hradecká 1,2, Vladimír Hönig2
Journal: Foods/MDPI
Dear Editor and Reviewers,
Attached please find the revised version of our manuscript foods-3039246, entitled “A Comprehensive Study for Determination of Free Fatty Acids in Selected Biological Materials: A Review” that we have revised according to the reviewers’ comments, and revised passages have been highlighted (in red) in the body text. The manuscript has been carefully checked and edited to avoid minor typo spell check.
On behalf of all authors, I would like to thank the referees and the editor for the suggestions and comments that helped us to improve the quality of our work substantially.
The detailed response to the reviewers’ comments and suggestions are as follows:
Reviewer #1:
Dear AUTHORS,
In my opinion, the review work would be scientifically valuable if it described the determination of fatty acids in biological materials using instrumental analytical methods, especially chromatographic methods: Ag-liqiud-chromatpgraphy, UPLC/HPLC-DAD/FLU/MS or GC with MS and MS/MS.
Lines 120-121, 223-236: FFAs profiles/concentrations were analysed in oil animal fat samples in alkaline solutions (i.e., after saponification !) Therefore: the title of the submitted manuscript misleads readers.
Your review on the determination of fatty acids using methods that also specify the procedures for hydrolysis of samples of animal fats or vegetable or fish oils would be more scientifically valuable. Conditions of hydrolysis methods (like temperature, duration) have a significant impact on the profile of mono- and poly-unsaturated fatty acids (especially conjugated fatty acids, e.g. in milk fats or pomegranate seed oils).
Therefore, I suggest the more interesting scientific scope and title of your manuscript: "A comprehensive study for determination of fatty acids in selected biologial materials: A Review" .
Response: We would like to thank the Reviewer for the comments. Discussions have been added to clarify this information.
In Page 1, lines 2-3:
The title of the article has been revised by considering the comments as ‘A Comprehensive Study for Determination of Free Fatty Acids in Selected Biological Materials: A Review’.
In Page 1, lines 16:
‘Biological materials’ has been written to catch the title.
In Page 1, lines 28-29:
‘the quality and composition of oils are critical factors, and’ added to catch the targeted reader.
In Page 3, lines 111-115:
The last paragraph of the Introduction has been revised for better understanding of the title and the review.
‘review aims to provide in-depth analysis of the methods used to determine fatty acids in biological materials, focusing on advanced instrumental analytical techniques. By expanding the scope to include alternative methods, this review seeks to offer a comprehensive guide for researchers and industry professionals in selecting the most appropriate methodologies for fatty acid analysis.’
In Page 7, lines 235-250:
Lines 223-236 became Lines 235-250 in the revised manuscript. Nothing changed, because we believe that we have created a great agreement with the Reviewer #1 by revising the title like they suggested.
In Page 11, lines 369-370 and 373:
Naming of sample types extended to present real objectivity.
In Page 11, lines 388-390:
An example research has been given for biological materials.
‘Schripsema et al. [99] investigated aqueous extracts of butter and margarine samples at 25 °C, with a spectral width of 10 ppm and a relaxation delay of 4 s, and the results were precise and comparable with GC and HPLC [99].
In Page 12-13, lines 427-457:
The section of Gas Chromatography has been improved by utilized samples and describing GC-MS/MS technique. The revised section is placed below.
‘Gas Chromatography (GC) is a spectroscopic method and a powerful technique for the analysis of volatile fatty acids. GC provides quantitative and effective results although it suffers presenting long analysis times [49]. Gas Chromatography -Mass Spectrometry (GC–MS), Flame Ionization Detector (GC-FID) and tandem Mass Spectrometry (GC-MS/MS) are common spectroscopic methods for the analysis of fatty acids in complex biological matrices such as oils (edible and harvest), animal fats, fish oils, dairy products and etc. This methodology does not require any sample pretreatment, making it highly convenient for straightforward application [85, 103]. For the FFA determination, the temperature range is foreseen from 40 °C to 240 °C, and the exact range depends on the applied sample [103].
GC-MS may detect more structural information, and present accurate databases for identification of fatty acids with higher efficiency and better selectivity than GC-FID [85]. However, lipid species requires high boiling point. Thus, labor-preparation is necessary for sample analysis in GC-MS to form and detect FFA of the triglycerides (TAGs), because all compounds should be volatile in GC-MS method, and it can cause to transform triglycerides (TAGs) to their relative FFAs due to utilizing high column temperature [48]. Moreover, GC-MS is capable to detect FFAs when they are neutralized by saponification, separation, and esterification [16]. Additionally, GC-MS/MS was found beneficial to define fatty acid profile of complex biological samples with strong structural identification capability by Chen et al. [104], Herrmann et al. [105] and others thanks to dual determination system. MS/MS application has not been confirmed for FFA detection of VO, whose content presents high amount of unsaturated C18, and has revealed as a suitable technique for detailed FFA determination of VO by Beneito-Cambra et al. [106]. This claim has paved the way of further investigations.
The utilized reagents have importance as much as methodology to obtain proper identification in GC-MS. Thus, Nzekoue et al. [107] exhibited an valuable research in this perspective and investigated salting-out agents which are NaCl, Na2SO4, NaH2PO4 and Na2CO3 to detect FFA (Figure 8). The solubility and ionic strength of NaH2PO4 were higher than other agents on the extraction of samples. NaCl, Na2SO4, and Na2CO3 were competed the analysis by decreasing their solubility. This developed method was useful to increase efficiency and detect short/mid FFA chains especially in lipids, ex, cheese [107].’
In Page 15, lines 493-506:
Ag+-HPLC has been described, and we have made sure agreeing with the comments about conjugated linoleic acid (CLA) in biological samples and hydrolysis method. The additional paragraph is given below.
‘One of the oldest ways to determine fatty acid profile of lipids is Ag+-HPLC whose column is silver-impregnated [112]. This method was described effective especially determination of conjugated linoleic acid (CLA) in biological samples which is a mixture isomer of linoleic acid containing a conjugated double bond and valuable ingredient in dietary supplements by Ostrowska et al. [112], Luna et al. [113] and Yurawecz et al. [114]. All samples has required methylation (FAME production) to detect isomers. Also, the study of Luna et al. proved that utilization of acidic catalyst (i.e. BF3 or HCl) and mild conditions (30-35 °C, 30 min.) are essential for accurate determination [113]. Moreover, hydrolysis of FAME might be required for the accurate FFA analysis of animal fats, VO and lipids regarding their complex structure [113, 115]. Hydrolysis parameters depends on tested material and utilized catalyst. Nevertheless, the previous researches showed that hydrolysis application as a preparation step of HPLC was helpful with specific temperature (60-90 °C) and duration time (30 min.-2 h) for accurate determination of poly-unsaturated fatty acids, glycerides and others which were adsorbed in the structure [113, 115, 116].’
In Page 15-16, lines 527 and 543-554:
HPLC-DAD has been mentioned in line 527 and described in line 543-554 regarding to Reviewer #1 comments. The relative paragraph for line 543-554 is placed below.
‘HPLC-DAD method was found beneficial for determination of phenolic compounds, whose accumulation in the oil might be problematic for human health like tyrosol, along with the official method of International Olive Oil Council (2009) by Ozbek et al. [123]. Moreover, fatty acid profile of seed oils were investigated with HPLC-DAD for evaluation of the quality of the oils as bioactive rich compounds (tocopherols, phytosterols and others) by Ozbek et al. [123] and Lyashenko et al. [124]. Yuenyong et al. [125] experimented effectiveness of HPLC-DAD over analysis of 50 different plant oils (0.5 g) which were diluted with dichloromethane until reaching the final volume of 1.00 mL. The solution was filtered (0.45 μm) before HPLC injection. The utilized mobile phases were methanol and water, operating in a gradient mode. The FFA determination results were obtained within 30 min., and they were precise and comparable with GC-MS results even application of pseudo-cereal oils such as cauliflower, leafy green seed oil and others [125].’
In Page 18-19, lines 608-631:
UPLC-Q/TOF-MS, fluorescence and UV techniques were exist in the first submitted manuscript. Thus, we believe this part is suitable according to comments of Reviewer #1.

Reviewer 2 Report
Comments and Suggestions for Authors This review work summarized the determination methods of free fatty acids in various sources. The synthesis is well-structured and -formatted, the language is fluent, and overall it is a acceptable. Some of my suggestions are as follows: Figure 2 is very important and relevant, and it is recommended that this be used as a basis for another table that prominently summarises the characteristics, strengths and weaknesses, and data variability of each method. This would be more informative for the reader When describing different determination samples, it is recommended that authors give advice on what methods are best for example, meat samples and what methods are best for cereal samples, so that the summary is more attractive to the reader. The conclusion could be a more concise and logical section
Author Response
Response to reviewers
Manuscript Number: foods-3039246
Title: A Comprehensive Study for Determination of Free Fatty Acids in Selected Biological Materials: A Review
Authors: Beyza Uçar 1,*, Zahra Gholami 1, Kateřina Svobodová 1,2,*, Ivana Hradecká 1,2, Vladimír Hönig2
Journal: Foods/MDPI
Dear Editor and Reviewers,
Attached please find the revised version of our manuscript foods-3039246, entitled “A Comprehensive Study for Determination of Free Fatty Acids in Selected Biological Materials: A Review” that we have revised according to the reviewers’ comments, and revised passages have been highlighted (in red) in the body text. The manuscript has been carefully checked and edited to avoid minor typo spell check.
On behalf of all authors, I would like to thank the referees and the editor for the suggestions and comments that helped us to improve the quality of our work substantially.
The detailed response to the reviewers’ comments and suggestions are as follows:
Reviewer #2:
This review work summarized the determination methods of free fatty acids in various sources. The synthesis is well-structured and -formatted, the language is fluent, and overall it is a acceptable. Some of my suggestions are as follows: Figure 2 is very important and relevant, and it is recommended that this be used as a basis for another table that prominently summarises the characteristics, strengths and weaknesses, and data variability of each method. This would be more informative for the reader When describing different determination samples, it is recommended that authors give advice on what methods are best for example, meat samples and what methods are best for cereal samples, so that the summary is more attractive to the reader. The conclusion could be a more concise and logical section
Response: We would like to thank the Reviewer for the comments. Discussions have been added to clarify this information.
In Page 20, lines 632-666:
‘3.Comparison of Different FFA Analysis Methods’ section has been added into the manuscript to make the review article more valuable like Reviewer #2 suggested. Recommendations were given regarding to the utilized sample for the analysis. The relative paragraphs of the section is placed below.
‘The utilized FFA analysis methods of biological materials were described in the previous sections. Among all FFA analysis methods given in Figure 2, classical titration method is a well-known and the oldest method for determination of overall FFA value. It presents high solvent consumption, large sample requirement and lack of accuracy. Thus, it created motivation for the improvement. Instrumental methods are developed as an alternative to classical titration method. Since this method is classified into different methodologies, the fundamental properties were given in Table 3 to provide a clear and concise comparison of the methods discussed. All these methods are applicable on FFA determination of oil samples as targeted.
Overall value of FFAs might be defined by colorimetric, thermometric titration and electrochemical methods. This leads inadequate FFA determination like classical titration method. Solvent consumption is moderated, and the solvent requirement is still high which makes them harmful for the environment. FIA and voltammetric methods share a potential of coupling with HPLC method, making them suitable and accurate for various applications [61, 65]. Nevertheless, voltammetric methods is highly effective for determination of heavy and toxic metals as itself [63].
Spectroscopic and chromatographic methods provide detailed information of FFA content which is crucial for understanding quality of oils. These methods require only minimal amount of solvent and sample, offering high accuracy, repeatability, reproducibility and sensitivity for determination of FFA (Table 3). Although the Raman method is feasible, its accuracy in FFA determination is limited due to overlapping peaks which might be improved by coupling it with FT [39, 89]. For unsaturated FFA determination, NMR and infrared methods are among the most accurate ones, as detailed in Table 1, and might also be used for solid sample analysis such as butter [99], cheese [134, 135] and meat [136] products. Besides, HPLC is a versatile alternative for determining fatty acid profile of various sample types, as it does not require high heat application. It has flexibility to employ with different methods such as florescence, FTIR and others to enhance accuracy and sensitivity. Ag+-HPLC is effective for lipids and unsaturated FFA determination with slightly higher solvent and sample requirement than NMR and infrared methods. HPLC-CAD is advantageous for FFA analysis of oil samples like VO while HPLC-DAD is preferable for analyzing phenolic compounds. Despite the requirement for volatile samples due to heat application, GC remains a well-known method. GC-MS/MS is particularly beneficial for analyzing the fatty acid profile of biological samples, such as cereal oils, due to its dual detectors, which prevent over estimation of weak signals [104, 105].’
In Page 21, lines 667:
The basis of Figure 2 has been taken and inserted into the generated Table which is ‘Table 3. Comparison of instrumental methods.’. This Table aimed to compare the described methods with solvent requirements, strengths, weaknesses and data variability of each method. The screen of table is placed below.
Table 3. Comparison of instrumental methods.
|
Instrumental Methods |
Solvent Requirement |
Data Variability |
Strengths |
Weaknesses |
||
|
Colorimetric |
Copper Soap Method |
Relatively High |
Overall value of FFAs |
Simplicity, low cost, reproducibility |
Limited determination on dark colored samples |
|
|
FIA Method |
Moderated |
Sensitivity, accuracy |
Stability Problem |
|||
|
Thermometric Titration Method |
Moderated |
Overall value of FFAs |
Simplicity, low cost, reproducibility |
Inadequate determination of FFA |
||
|
Electrochemical |
Voltammetric |
Moderated |
Overall value of FFAs |
Low cost |
Overlapping and reproducibility problems |
|
|
EC Method |
Low cost |
Accuracy and reproducibility of the method mainly depend on the conditions of the electrodes. |
||||
|
pH Metric Method |
Low cost, accuracy, repeatability, reproducibility, sensitivity |
Inadequate determination of FFA |
||||
|
Spectroscopic |
Infrared Method |
Low |
In-depth understanding |
Non-destructive, accuracy, repeatability, reproducibility, sensitivity |
Sample Preparation is required. |
|
|
Raman Method |
Overlapping |
|||||
|
NMR Method |
Expensive |
|||||
|
Chromatographic |
GC |
Low |
In-depth understanding |
Feasible, accuracy, repeatability, reproducibility sensitivity |
Sample Preparation is required. |
|
|
HPLC |
Sample Preparation is required. Coupling with other methods could be required for sensitivity and accuracy. |
|||||
In Page 22, lines 671-715:
The conclusion has been revised to create a more concise and logical section through the revised manuscript. The methods were compared precisely to make this section more attractive and informative to the reader regarding the comments of Reviewer #2. The whole ‘4. Comparisons’ section is placed below.
‘The determination of free fatty acids (FFAs) is critical for assessing the quality of oils used in various industries. This review highlights the strengths and weaknesses of different FFA determination methods, including classical titration, chromatographic techniques, and spectroscopic methods.
The classical titration method is highly toxic due to its substantial solvent requirement. Even though colorimetric, electrochemical and thermometric methods have less solvent consumption than classical titration method except copper soap and EC methods, they still rely on solvents, thereby lacking an eco-friendly approach.
Colorimetric, electrochemical and thermometric methods offer faster results than spectroscopic methods. Trained personal, high maintenance cost and calibration does not be required for their application. The main weaknesses of these methods are high solvent usage, low frequency, stability obstacles, overlapping on the results, accuracy and reproducibility problems, especially for methods like copper soap, FIA, Voltammetric, EC which pose significant drawbacks. The latter, they only provide total FFA value instead of detailed chemical compositions.
Whereas, spectroscopic (infrared, Raman, NMR and chromatographic) methods provide rapid and non-destructive analysis, making them suitable for a wide range of applications such as analyzing of oil and lipid samples. The main disadvantages are the need of sophisticated equipment and expertise. Nevertheless, infrared, Raman and chromatographic methods require sample preparation unlike NMR. Unless instrumental cost is an obstacle for usage of NMR, it stands out as a promising method due to its excellent analysis results and minimal weakness. Raman spectroscopy offers well promising results if the issue of overlapping and low intensity peak determination overcome. Especially NIR spectroscopy takes attention for being non-destructive despite of color dependency of the applied sample. Among all chromatographic methods, HPLC-DAD, Ag+-HPLC and GC-MS/MS offer high sensitivity and detailed analysis for phenolic compounds and biological samples (i.e. cereal oils) while UPLC, HPLC-CAD, and GC-MS methods are more precise for edible oil analysis.
Fast FFA determination kits like 3M TM oil quality test, strips, FASafe and CDR FoodLabR are placed as alternative rapid test methods to detect FFA value. Inherently, these fast determination methods suffer from the same limitations as other methodologies except for spectroscopic. Thus, fluorescence and UV spectroscopies take attention to provide alternative approach for rapid determination of FFAs. Although they are not capable to detect wide range compounds on their own, they can be effectively combined with other methods like FTIR and HPLC to enhance sensitivity in analyzing oil and lipid samples.
The suitable FFA determination method is demanded to present accuracy, high sensitivity, repeatability and reproducibility, and it is determined by utilized eco- and user- friendly detection system, applied samples, purpose of work, requirement of trained workers. This study suggest that combining methods, such as FIA-HPLC, Fluorescence-FTIR or Raman-FT, or coupling with enhanced detectors using improved solvent types, can overcome current weaknesses. Future research should focus on developing eco-friendly and cost-effective methodologies that maintain high accuracy and reproducibility. By selecting the appropriate method based on sample type and analysis requirements, researchers can ensure precise and reliable FFA determination.’

Round 2
Reviewer 1 Report
Comments and Suggestions for Authors
Dear Authors,
In my opinion, the description of Ag-HPLC-DAD-methods for determining CLA isomers and conjugated trienes isomers (like CLnA isomers) needs to be improved. (Conjugated triens are very important fatty acids - see J. Anim. Feed Sci., 26, 2017, 3–17; https://doi.org/10.22358/jafs/68862/2017). Thus, please provide information on the possibility of distinguishing geometric and positional isomers of CLA using the Ag-HPLC-DAD method (see the paper: J. Anim. Feed Sci., 2003, 12, 369-382; DOI: https://doi.org/10.22358/jafs/67717/2003).
Moreover, please provide information on the original internal standards (IS) used in analysis of CLA and/or conjugated trienes isomers by Ag-HPLC-DAD (Czech J. Anim. Sci., 2011, 56(1), 23-29; DOI:10.17221/336/2009-CJAS).
Furthermore, the method of choise of conjugated isomers (like CLA, CLnA; > isomers) is Ag-HPLC/UPLC-DAD method, obviously without pre-column derivatization (due to high chemical instability).
Saponification of biological materials requires mild conditions. This is contrary to the recommendation presented in the manuscript (see lines 509-512). Hydrolysis: specific temperature (60-90 °C) and duration time: 30 min.-2 h (see lines 510-511). Pre-column hydrolysis under these conditions can be used to analyze saturated (SFA) and mono-unsaturated fatty acids (MUFA).
Especially mild condition of saponification applies to samples containing CLA isomers, conjugated trienes isomers (like CLnA) and LPUFA (like EPA, DPA or DHA). This is very important in the pre-column preparation of analyzed biological samples. Therefore, please provide the main conclusions presented in the paper: "Improved saponification then mild base and acid-catalyzed methylation is a useful method for quantifying fatty acids, with special emphasis on conjugated dienes. Acta Chromatogr., 2007, 18, 59-71.
Author Response
Response to reviewers
Manuscript Number: foods-3039246
Title: A Comprehensive Study for Determination of Free Fatty Acids in Selected Biological Materials: A Review
Authors: Beyza Uçar 1,*, Zahra Gholami 1, Kateřina Svobodová 1,2,*, Ivana Hradecká 1,2, Vladimír Hönig2
Journal: Foods/MDPI
Dear Editor and Reviewers,
Attached please find the revised version of our manuscript foods-3039246, entitled “A Comprehensive Study for Determination of Free Fatty Acids in Selected Biological Materials: A Review” that we have revised according to the reviewers’ comments, and revised passages have been highlighted (in red) in the body text. The manuscript has been carefully checked and edited to avoid minor typo spell check.
On behalf of all authors, I would like to thank the referees and the editor for the suggestions and comments that helped us to improve the quality of our work substantially.
The detailed response to the reviewers’ comments and suggestions are as follows:
Reviewer #1:
Dear Authors,
In my opinion, the description of Ag-HPLC-DAD-methods for determining CLA isomers and conjugated trienes isomers (like CLnA isomers) needs to be improved. (Conjugated triens are very important fatty acids - see J. Anim. Feed Sci., 26, 2017, 3–17; https://doi.org/10.22358/jafs/68862/2017). Thus, please provide information on the possibility of distinguishing geometric and positional isomers of CLA using the Ag-HPLC-DAD method (see the paper: J. Anim. Feed Sci., 2003, 12, 369-382; DOI: https://doi.org/10.22358/jafs/67717/2003).
Moreover, please provide information on the original internal standards (IS) used in analysis of CLA and/or conjugated trienes isomers by Ag-HPLC-DAD (Czech J. Anim. Sci., 2011, 56(1), 23-29; DOI:10.17221/336/2009-CJAS).
Furthermore, the method of choise of conjugated isomers (like CLA, CLnA; > isomers) is Ag-HPLC/UPLC-DAD method, obviously without pre-column derivatization (due to high chemical instability).
Saponification of biological materials requires mild conditions. This is contrary to the recommendation presented in the manuscript (see lines 509-512). Hydrolysis: specific temperature (60-90 °C) and duration time: 30 min.-2 h (see lines 510-511). Pre-column hydrolysis under these conditions can be used to analyze saturated (SFA) and mono-unsaturated fatty acids (MUFA).
Especially mild condition of saponification applies to samples containing CLA isomers, conjugated trienes isomers (like CLnA) and LPUFA (like EPA, DPA or DHA). This is very important in the pre-column preparation of analyzed biological samples. Therefore, please provide the main conclusions presented in the paper: "Improved saponification then mild base and acid-catalyzed methylation is a useful method for quantifying fatty acids, with special emphasis on conjugated dienes. Acta Chromatogr., 2007, 18, 59-71.
Response: We would like to thank the Reviewer for the comments. Discussions have been added to clarify this information.
In Page 3:line 118, Page 6:line 201, Page 7:line 245 and 263, Page 9:line 334, Page 10:line 345, Page 11:line 358 and 360, Page12:line 418, Page 13:line 454, Page 14:line 472, Page 15:line 479, 481, 482 and 504, Page 18:line 619, 622, 624, 626 and 628, Page 19:line 634, Page 20:line 647 and 655:
We have realized that cross-references for Figures and Tables disappeared from text in the template. Thus, we wrote manually in red.
In Page 16: line 513:
We added the name of Ag+ into the sentence (511-513). The sentence is placed below.
‘Moreover, HPLC system can be employed with various alternative systems, most commonly evaporative light scattering detector (ELSD), charged aerosol detector (CAD), diode array detector (DAD), UV, MS and Ag+ for FFA detection of biological samples.’
In Page 16 and 17: lines 542-586:
Ag+-HPLC methodology carried under to HPLC-DAD methodology to create better connection corresponding to the recommend of Reviewer #1 about Ag+-HPLC-DAD methodology. This part improved like Reviewer #1 suggested, and the revisions explained below.
Lines 543-553;
The description of Ag+-HPLC methodology and CLA isomers extended with the recommended references (https://doi.org/10.22358/jafs/67717/2003, https://doi.org/10.22358/jafs/68862/2017).
Lines 555-559;
Previously given parameters for methylation have been deleted from the text. Mild condition saponification and main conclusions of suggested article (Acta Chromatogr., 2007, 18, 59-71) considered.
Lines 561-565;
Hydrolysis parameters, specific temperature (60-90 °C) and duration time: 30 min.-2 h, have not been changed because these parameters covered hydrolysis conditions of suggested articles which are explained in following lines (565-586).
Lines 565-576;
The possibility of distinguishing geometric and positional isomers of CLA by the Ag+-HPLC-DAD method were explained through recommended article (https://doi.org/10.22358/jafs/67717/2003).
Lines 576-586;
The usage of original internal standards (IS) for CLA analysis by Ag+-HPLC-DAD-UV was demonstrated with utilization of mentioned article (Czech J. Anim. Sci., 2011, 56(1), 23-29; DOI:10.17221/336/2009-CJAS).
The all paragraph is placed below with the revisions, in lines 542-586.
‘One of the oldest ways to determine fatty acid profile of lipids is Ag+-HPLC whose column is silver-impregnated [121]. This silver ions form polar complexes with unsatu-rated centers of fatty acids reversibly, though the residual silanol groups of the column creates interaction with carboxylic acid groups of FAs [122]. This method was described effective especially determination of conjugated linoleic acid (CLA) which is a mixture of linoleic acid isomers, especially cis-9,trans-11 C18:2 (c-9,t-11 C18:2), in biological samples like food products from ruminant origin and some plant products, containing a conjugat-ed double bond and valuable ingredient in dietary supplements by Ostrowska et al. [121], Luna et al. [123], Yurawecz et al. [124] and Białek et al. [125] without derivatization. Moreover, CLA isomers display beneficial effect on arranging fatty acid composition of blood, antidiabetic and anticancerogenic properties with limited daily usage (2-3 g per day) due to ability of nutrient adsorption in humans [125]. All samples has required methylation (FAME production) to detect isomers. The study of Luna et al. proved that uti-lization of acidic catalyst (i.e. BF3 or HCl) and mild conditions are essential for accurate determination [123]. Czauderna et al. has various researches on determination of CLA by Ag+-HPLC, and one of their researches pointed out that the combination of mild saponifi-cation with base and acid catalyzed methylation at 40°C for 1 h prevented further isomer-ization during sample analysis [126] Moreover, hydrolysis method might be required for the accurate FFA analysis of animal fats, VO and lipids regarding their complex structure [123, 127]. Hydrolysis parameters depends on tested material and utilized catalyst. Nev-ertheless, the previous researches showed that hydrolysis application as a preparation step of HPLC was helpful with specific temperature (60-90 °C) and duration time (30 min.-2 h) for accurate determination of poly-unsaturated fatty acids, glycerides and others which were adsorbed in the structure [122, 123, 127, 128]. Additionally, Czauderna et al. [122] combined Ag+-HPLC with DAD which operated with a UV range from 195 to 400 nm with 1.2 nm spectral resolution and 1 spectrum per sec. measurement frequency to define CLA isomers. They prepared n-hexane/acetic acid/acetonitrile (98.4:1.6:0.0125 v/v/v) solution as mobile phase and equilibrated the column 40 min. before sample injection. Then, biological samples prepared by hydrolysis method with 2 M NaOH (3.5-4.0 mL) at 80-85°C for 30-35 min. followed by acidification with 4 M HCl to pH ~2, extraction of FFAs with 3.5 mL of dichloromethane, drying with Na2SO4 (50-100 mg) and solvent removal under Ar. They confirmed this altered system had great agreement with separation of ge-ometric and positional isomer mixtures e.g. trans-trans, cis-trans, trans-cis and cis-cis configurations in narrow UV range (228-240 nm). CLA isomers were particularly detected at 234 nm [122]. In another research of Czauderna et al. [129], the original internal stand-ards (IS) used in analysis of CLA and/or conjugated trienes isomers by Ag-HPLC-DAD-UV. Biological samples prepared with 1 mL 1M KOH in methanol, 1 mL 2M KOH in water and 25 μL IS (6 mg sorbic acid (c2c4C6:2) in 15 mL chloroform), followed by flushing with Ar stream for 1 min. and left overnight at room temperature for hydroly-sis. Then, 1.5 mL of water were added and the solution was acidified with 6 M HCl to pH ≈ 2 to produce FAME. The latter, this sample extracted with 3.5 ml of dichloromethane, dried with Na2SO4 (50-100 mg) and connected to Ar stream for solvent removal. As a result, CLA isomers were detected at 234 nm like in previous research [122], and IS assisted to removal of impurities from mobile phase and altering baseline noise for better detection, was ob-served at 259 nm [129].’
In Page 23:lines 727-728;
Ag+-HPLC-DAD-UV name mentioned.
In Page 28:lines 997-1015:
The relative references for Ag+-HPLC methodology marked as red, because the place of the references was changed besides additional references.
Reviewer 2 Report
Comments and Suggestions for Authors
It could be accepted
Author Response
Response to reviewers
Manuscript Number: foods-3039246
Title: A Comprehensive Study for Determination of Free Fatty Acids in Selected Biological Materials: A Review
Authors: Beyza Uçar 1,*, Zahra Gholami 1, Kateřina Svobodová 1,2,*, Ivana Hradecká 1,2, Vladimír Hönig2
Journal: Foods/MDPI
Dear Editor and Reviewers,
Attached please find the revised version of our manuscript foods-3039246, entitled “A Comprehensive Study for Determination of Free Fatty Acids in Selected Biological Materials: A Review” that we have revised according to the reviewers’ comments, and revised passages have been highlighted (in red) in the body text. The manuscript has been carefully checked and edited to avoid minor typo spell check.
On behalf of all authors, I would like to thank the referees and the editor for the suggestions and comments that helped us to improve the quality of our work substantially.
The detailed response to the reviewers’ comments and suggestions are as follows:
Reviewer #2:
It could be accepted.
Response: We would like to thank the Reviewer for the acceptance of the paper.